# Enhancing pigment production by a chromogenic bacterium (*Exiguobacterium aurantiacum*) using tomato waste extract: A statistical approach

Birhanu Zeleke[1], Diriba Muleta [2]*, Hunduma Dinka[1], Dereje Tsegaye[3], Jemal Hassen [4]

1 Department of Applied Biology, School of Applied Natural Science, Adama Science and Technology University, Adama, Oromia, Ethiopia, 2 Environmental Biotechnology Unit, Institute of Biotechnology, Addis Ababa University, Addis Ababa, Ethiopia, 3 Department of Applied Chemistry, School of Applied Natural Science, Adama Science and Technology University, Adama, Oromia, Ethiopia, 4 Adama Public Health Research and Referral Laboratory Center, Adama, Oromia, Ethiopia

* dmuleta@gmail.com

## Abstract

There is a high demand for microbial pigments as a promising alternative for synthetic pigments, primarily for safety and economic reasons. This study aimed at the optimization of yellowish-orange pigment production by *Exiguobacterium aurantiacum* using agro-waste extracts as a growth substrate. Air samples were collected using the depositional method. Pure cultures of pigment producing bacteria were isolated by subsequent culturing on fresh nutrient agar medium. The potent isolate was identified using MALDI-TOF technique. Screening of culture conditions was done via Plackett-Burman design that highlighted culture agitation rate, initial medium pH, and yeast extract concentration as the most significant variables ($p < 0.0001$) in influencing pigment production with further optimization step using response surface methodology. Among the tested agro-waste decoctions, tomato waste extract was selected for fermentation due to higher optical density of the isolate when cultivated in it compared to the other agro-waste extracts. Under optimized conditions, 0.96 g/L of pigment was extracted from 4.73 g/L of culture biomass, representing a 1.6-fold increase compared to un-optimized conditions. Spectroscopic and chromatographic analyses confirmed the presence of various functional groups, with carotenoids identified as the primary compounds responsible for the yellowish-orange pigmentation. These findings demonstrate the feasibility of enhancing bacterial pigment production using agro-waste substrates, highlighting its potential for large-scale industrial applications.

## 1. Introduction

The growing demand for colored products has led to a proportional increase in the use of synthetic dyes, despite their environmental hazards. Since the mid-19th

**Data availability statement:** All relevant data are within the manuscript.

**Funding:** The author(s) received no specific funding for this work.

**Competing interests:** The authors have declared that no competing interests exist.

century, synthetic dyes have largely replaced natural pigments due to their lower production costs. The synthetic dyes are continued to be extensively used regardless of their deleterious effects [1,2]. It is estimated that nearly 800,000 tons of synthetic dyes are produced annually worldwide, with more than 10,000 different dyes in use [3].

Regardless of the dye type, the final operations of all the dyeing processes require washing to remove unfixed dyes. However, the continuous use of synthetic dyes and the discharge of dyed effluents pose serious environmental challenges. Due to their resistance to light, temperature, water, detergents, and chemicals, these effluents persist in the environment for long periods [4]. The heavily dependence on synthetic dyes, coupled with huge amounts of water consumption, disrupts ecological balance by diminishing productivity and threatening biodiversity.

Environmental pollution and biodiversity conservation concerns call for eco-friendly alternatives to synthetic dyes, with microbial pigments that have emerged as promising substitutes. Microorganisms produce various pigments through their bioactive compounds, offering promising possibilities for several applications [5].

In view of the possible commercial values of microbial pigments, yield maximization and production cost minimization are key parameters to make the production process economically viable [6,7]. Utilizing low-cost agro-wastes that are rich in cellulose, hemicellulose, lignin, organic acids, salts, and minerals offer a sustainable approach to maximize microbial pigment production while supporting renewable resource utilization [8].

In optimization experiments, the primary objective is to identify factors that significantly influence pigment yield to determine their optimal values [9]. Plackett-Burman Design (PBD) is an effective statistical tool for screening process variables by allowing the selection of key influential factors from a large set of variables, which can then be fixed for further optimization steps [10].

PBD has been successfully employed in various applications, including the screening of process components affecting enzyme production by *Bacillus* species from dairy effluent [11], in identifying trace nutrients essential for glycolipopeptide bio-surfactant production [12], in evaluating process variables in the production of gedunin-loaded liposomes [13], and in the optimization of industrial-grade media to enhance the biomass production of *Weissella cibaria* [14].

Response surface methodology (RSM) enhances pigment production by reducing process variables, time, and cost. It effectively optimizes fermentation parameters by analyzing variable interactions and constructing a precise mathematical model of the process [15]. RSM has been applied to enhance lysine–methionine biosynthesis by *Pediococcus pentosaceus* RF-1 [16], improve carotenoid pigment yields in *Cellulosimicrobium* strain AZ [17], optimize industrial media for *Weissella cibaria* [14], evaluate the thermal stability of *Monascus purpureus* pigments [18], boost biomass production in psychrotolerant *Paenibacillus* species BPW19 [19], and facilitate yellowish-orange pigment synthesis in *Chryseobacterium artocarpi* CECT 8497 [20].

This study aimed to optimize culture conditions and media components for *Exiguobacterium aurantiacum* pigment production using PBD and RSM, while evaluating

agro-waste extracts as cultivation substrates. The study successfully identified optimal culture parameters and medium concentrations that enhanced pigment yield, with experimental validation confirming the results.

## 2. Materials and methods

### 2.1. Chemicals, equipment and reagents

Nutrient agar (NA) medium was used to support colony growth, while liquid agro-waste extracts were used for cultivation. Pigment extraction was performed using analytical-grade organic solvents from Adama Public Health Research and Referral Laboratory Center, with centrifugation that aided separation. Absorbance measurements of dependent variables were obtained using a Palin test Wagtech Potalab+ photometer (UK) and a VWR P9 UV-Vis spectrophotometer (China). Aseptic conditions were rigorously maintained throughout the experimental process.

### 2.2. Sample collection, isolation and identification of pigment- producing isolates

Air samples were collected using the depositional method according to [21], where nutrient agar plates (Oxoid) were exposed to ambient air for 15 minutes and then incubated at 30°C for 48 hours. The plates were then incubated at 30°C for 48 hrs. Pure cultures of pigment-producing isolates were obtained through successive culturing on fresh nutrient agar. The most potent pigment producers were screened and identified via MALDI-TOF mass spectrometry, following the guidelines provided by Bruker Daltonics for the Microflex LT Biotyper system [22].

A colony of the pigmented isolate was suspended in 300 μL of ultrapure water using a sterile micropipette tip, followed by the addition of 900 μL of absolute ethanol and thorough mixing. The mixture was centrifuged at 12,000 rpm for 3 minutes, and the supernatant was discarded. After air drying the precipitate, 20 μL of 70% formic acid and 20 μL of acetonitrile were added and mixed well. A second centrifugation at 5,000 rpm for 3 minutes separated the supernatant, which was transferred to a new Eppendorf tube. Finally, after air drying, a microliter of the extracted supernatant was applied to the target plate and covered with one microliter of α-Cyano-4-hydroxycinnamic acid (CHCA) matrix solution for analysis.

Mass spectral fingerprints were obtained using the EXS3000 MALDI-TOF-MS (Bruker Daltonics, Germany) following its operational manual. The system was set at a laser frequency of 60 Hz, with an ion source voltage of 1.8 kV and a lens voltage of 6 kV. Spectra were recorded across a mass-to-charge ratio (m/z) range of 2000–20,000. *E. coli* (ATCC25922) served as the positive quality control, while α-CHCA was used as the negative control.

The mass spectra of the isolate were analyzed against bacterial spectra in the MALDI Biotyper library (MBL), which houses species-specific fingerprints for diverse bacterial taxa. This comparison helped to determine the isolate's taxonomic classification. Identification confidence was scored as follows: values below 1.69 indicated an unreliable genus match, scores between 1.70 and 1.99 suggested a probable genus, and scores exceeding 2.00 confirmed a probable species assignment [23].

### 2.3. Experimental design for screening and optimization of growth conditions

**2.3.1. Cultivation of cultures on agro-waste extracts (AWEs).** The growth of pigment-producing isolates was assessed *in vitro* using growth curve analysis [24] in twelve locally available agro-waste extracts (AWEs) that included potato, cabbage, tomato, orange, cannon ball cabbage, onion, watermelon, papaya, carrot, banana, beetroot peels, and bread leftovers. Fruit and vegetable waste extracts are nutritionally rich, containing dietary fibers, proteins, carbohydrates, and essential minerals such as potassium, calcium, and magnesium, along with vitamins A, C, and E [25–28].

Agro-wastes were sorted at the source, thoroughly washed, sun-dried, ground into powder, and sieved using a stainless steel sieve (1–2 mm mesh). Extracts were obtained through a modified decoction method according to [29], where 50 g of ground agro-waste residues were boiled in a liter of distilled water contained in 2 L capacity of flasks. After reaching boiling point, the mixture was simmered for 45 minutes, then strained through gauze fabric and filtered using Whatman

No.1 filter paper. The extracts were stored in sterilized, stoppered Schott bottles, autoclaved, and refrigerated for successive fermentations as growth substrates for the selected bacterium.

The inoculum of the potent isolate (PPPI-6) was prepared using the shake flask culture method, where a loopful of culture was transferred to nutrient broth and incubated over night at 30°C for activation. A pre-culture of 10 mL one-day-old culture suspension of the isolate was cultivated in each AWEs to assess their effectiveness in supporting culture growth. Growth was monitored daily by measuring optical density at 600 nm ($OD_{600}$) until the stationary phase. At the stationary phase, the $OD_{600\,nm}$ values of each broth culture were computed and statistically analyzed to identify the most nutritious AWEs that enhanced growth compared to the nutrient broth (control).

Standardization was employed using PBD to screen significant factors that affect pigment production. RSM was used to fine-tune those significant factors for maximum pigment yield, and statistical models predicted the best conditions for enhanced culture growth and pigment production using the best AWE that supported the culture growth as an alternative low-cost substrate. All the experiments were performed in triplicates and average values were used for analysis.

**2.3.2. Screening of process variables using PBD.** Fermentation was carried out in 250 mL flasks with a working volume of 150 mL nutrient broth as the basal medium to assess the effects of various variables to determine the key factors influencing the process. Screening of culture conditions and nutrient concentrations were performed using the Plackett-Burman experimental design, a fractional factorial method that identifies the most significant process variables that affect bacterial growth [10].

Nine variables comprising of incubation temperature, initial medium pH, culture agitation rate, incubation period, inoculum age, inoculum size, glucose concentration, yeast extract concentration, and salt concentration were assessed for their influence on culture growth. Their selection was guided by extensive literature review and laboratory trials. Each variable was tested at two levels, low (-) and high (+), across 12 randomized experimental runs (Table 1), following the Plackett-Burman experimental design and analyzed using Design Expert Statistical Software (version 13).

To minimize variability and ensure a reliable assessment of the studied variables, the average culture $OD_{600\,nm}$ values were analyzed. The main effects model and Pareto analysis helped pinpoint the most influential factors affecting the response. Based on ANOVA results, the three most significant variables impacting culture growth were selected for further optimization.

**2.3.3. Optimization of the levels of significant factors using RSM.** RSM with a face-centered central composite design (CCD) was applied to develop a quadratic model for response variables, estimating first and second-order terms using a three-level matrix (Table 2). Non-significant variables (incubation temperature, inoculum age, inoculum size, NaCl, and glucose concentrations) were maintained at average values [30,31]. Since maximum pigment production

**Table 1. Independent variables selected for screening using PBD.**

| S. No | Variable | Unit | Experimental level | |
|---|---|---|---|---|
| | | | Low (−1) | High (+1) |
| 1 | Incubation temperature | ° C | 23 | 37 |
| 2 | pH | | 5 | 10 |
| 3 | Culture agitation rate | rpm | 60 | 180 |
| 4 | Incubation period | hr | 24 | 72 |
| 5 | Inoculum size | % | 1 | 2.5 |
| 6 | Inoculum age | hr | 12 | 36 |
| 7 | Yeast extract | % | 0.1 | 1 |
| 8 | Glucose | % | 0.5 | 1.5 |
| 9 | Salt (NaCl) | % | 5 | 10 |

**Table 2. Design matrix of significant variables for optimization with their corresponding levels.**

| Variable name | Units | Low (−1) | High (+1) | -α | +α |
|---|---|---|---|---|---|
| Culture agitation rate ($X_1$) | rpm | 60 | 180 | 60 | 180 |
| pH ($X_2$) | | 5 | 10 | 5 | 10 |
| Yeast extract ($X_3$) | % | 0.1 | 1.0 | 0.1 | 1.0 |

aligns with the stationary growth phase, bacterial cultures were incubated until reaching this phase by monitoring turbidity measurement at 600 nm [32]. Model assumptions and fitness were evaluated to ensure statistical reliability.

Graphical optimization was employed to visualize process variable interactions by enabling the identification of optimal parameter combinations using surface plots [33]. Additionally, the desirability function method was applied for simultaneous response optimization that determines the best operating conditions for maximum efficiency. Factor settings were adjusted to maximize overall desirability, with input variables set "in range" to establish ideal growth conditions and output variables set to "maximize" for optimal pigment production [34].

## 2.4. Verification test

Before optimization, culture cultivation was performed using tomato waste extract (TWE) as the substrate while maintaining all the growth conditions at average values with initial pH (7.5), agitation rate (120 rpm), incubation temperature (30°C), inoculum age (24 hours), inoculum size (2%), NaCl (7.5%), glucose (1%), and yeast extract (0.55%). This test allowed for an assessment of the significant variables influencing bacterial growth and pigment production [30,31].

To validate the optimized conditions, cultivation was performed in 150 mL of freshly prepared TWE under predicted optimal parameters. Bacterial growth kinetics was monitored via $OD_{600\,nm}$ measurements by withdrawing a 10 mL of culture broth and replacing with an equal volume of sterile TWE. Growth dynamics was plotted over time. At the stationary phase, the culture was centrifuged using a ROTINA 380 R bench top centrifuge (Hettick, Germany) to collect cell biomass for crude pigment extraction. Experiments were conducted in triplicates, with average response values used to ensure reliable estimates of the optimized factors.

## 2.5. Pigment extraction

Pigment extraction was carried out using the solvent extraction method according to [19]. After centrifugation, the supernatant was discarded, and the collected cell biomass was weighed and re-suspended in various organic solvents that included methanol, ethanol, acetone, and chloroform. The most effective solvent for pigment extraction was identified by scanning its absorbance across a wavelength range of 350–750 nm to determine the maximum absorption wavelength (λmax), which represents the pigment's optimal light absorption. To ensure accuracy, spectrophotometric scans for each solvent were repeated, and the solvent yielding the highest absorbance at λmax was selected as the most efficient for dissolving the pigment.

Following re-suspension in selected solvents, culture suspensions were vortexed for 10 minutes, heated at 60°C for 30 minutes in a water bath, and acidified with 3N HCl to enhance pigment extraction [35,36]. The treated broth was then subjected to repeated centrifugation at 5000 rpm and 4°C for 20 minutes until the residue turned white. The residue was discarded, and the colored supernatant was filtered through a 0.45 μm filter paper. Finally, the extract was stored in a biosafety cabinet (BSC) for a week to allow solvent evaporation.

The dry biomass and crude pigment extract were quantified using the gravimetric technique [37], where biomass weight was calculated based on mass changes. After solvent evaporation, the crude pigment's weight was determined and further analyzed by measuring its absorbance across the 350–750 nm wavelength range using a UV-Vis spectrophotometer [38].

## 2.6. Pigment characterization

The extracted pigments were filtered and concentrated to remove impurities and enhance its detectability, respectively. Various analytical techniques were applied to characterize the pigment focusing on its chemical composition, spectroscopic properties, and chromatographic separation for identification and structural elucidation.

### 2.6.1. Infrared (IR) spectroscopy.

The Attenuated Total Reflectance-Fourier Transform Infrared (ATR-FTIR) spectroscopy technique was employed for spectrum analysis using a Thermo Scientific Nicolet iS50 FTIR spectrometer (USA) equipped with an ATR device. Infrared light passed through the crystal, where it was partially absorbed by the sample pressed against it. The reflected light traveled back through the crystal and reached the FTIR detector [39], enabling the identification of functional groups present in the pigment.

Two milligrams of finely ground pigmented extract were directly placed onto the ATR crystal, ensuring optimal contact under 700 kg/cm² pressure. Infrared radiation from the spectrometer was directed onto the ATR crystal, with the output focused on a deuterated triglycine sulfate (DTGS) detector coated with potassium bromide. The spectrum was collected across the 4000–400 cm$^{-1}$ range, averaging 32 scans at a resolution of 16 cm$^{-1}$ [40]. The resulting spectral data were then analyzed to determine the sample's molecular composition.

### 2.6.2. UV-Visible spectroscopy.

UV-Visible spectroscopy was employed to identify the extracted pigment based on its characteristic absorption peaks. The pigmented extract was dissolved in 99.5% methanol, with 10 mL of the solution placed into a clean cuvette and analyzed using a P9 double-beam UV-visible spectrophotometer (VWR, China), with methanol serving as the negative control. The absorbance values of the sample were adjusted to between 0.1 and 1.0 absorbance units through dilution. Absorbance measurements were conducted across the 350–750 nm wavelength range to determine the pigment's peak absorption. To ensure accuracy and consistency, readings were taken in triplicates.

### 2.6.3. Chromatographic analysis.

The pigment was analyzed using an Agilent 1260 Infinity II LC-MS 6495 system (Germany), equipped with electrospray ionization (ESI) source and a Triple Quadrupole mass analyzer to separate and identify individual components based on their mass-to-charge ratio [41]. The sample was dissolved in 99.5% methanol and centrifuged at 10,000 rpm for 10 minutes at 4°C. The clear supernatant was transferred into an auto-sampler vial, from which 10 µL was injected into the liquid chromatography (LC) system. Chromatographic separation was performed using a standard reversed-phase C18 column (10 cm × 4.6 mm, 3 µm) at 35°C, followed by mass spectral detection.

Once separated, the sample entered the mass spectrometer for ionization, facilitated by nitrogen gas for efficient spray ionization. Instrument settings included a capillary voltage of 3000 V, a gas flow rate of 5 L/min, a gas temperature of 300°C, and a nebulizer pressure of 50 psi to optimize ion formation and analysis.

A 40-minute gradient elution program was used for effective separation and ionization by employing 0.1% acetic acid in water and methanol at a 1 mL/min flow rate. The elution started with 90% water-acetic acid and 10% methanol, with methanol gradually increasing from 10% to 90% over 35 minutes, remaining constant for the final 5 minutes before returning to the initial conditions. Spectra were scanned across a mass-to-charge (m/z) ratio of 50–1000, and the generated spectra were compared with a reference database to identify chromophoric compounds present in the pigment extract.

## 2.7. Data analysis

Design Expert software (Version 13) was utilized for variable screening and optimization, while all the AWEs screening experiments were conducted in triplicates and analyzed using Microsoft Excel. Linear regression analysis assessed the relationship between independent and dependent variables, and a One-way ANOVA test compared the means across different input variables to identify significant variations.

The P-value was calculated by evaluating variance between and within groups at a significance level ($\alpha$) < 0.05 that serves as the threshold for determining statistically significant differences [16,42]. Post-hoc analysis using the Bonferroni correction method was applied to pinpoint specific group differences.

## 3. Results and discussion

### 3.1. Isolation and identification of pigment producing bacteria

Pigmented bacterial colonies were successfully retrieved from air samples incubated at 30°C on nutrient agar medium after 48 hours. The isolation process involved streaking the mixed colonies on a pre-solidified agar surface to obtain pure, single colonies, each representing an identical strain derived from a single progenitor cell (Fig 1). The distinct morphology and clear separation of colonies confirmed successful isolation that enabled further analysis.

Among the isolates, PPPI-6, identified as *Exiguobacterium aurantiacum* via MALDI-TOF mass spectrometry (MS) was selected based on non-pathogenicity, supported by literature review [43], rapid growth in the employed agro-waste extract, better pigment yield compared to other recovered pigment-producing isolates, and pigment stability under various environmental conditions in repeated laboratory trials.

The MALDI-TOF-MS spectra in Fig 2 illustrate distinct mass-to-charge ratio (m/z) peaks associated with bacterial proteins, serving as a unique fingerprint for identifying and classifying the pigment-producing strain.

These characteristic peak patterns highlight specific biomolecules involved in pigment production, validating the bacterial identity and underscoring its potential industrial applications, such as natural pigment synthesis.

### 3.2. Culture cultivation

Three agro-waste extracts of banana peels, beetroot peels, and bread leftovers were unable to support bacterial culture growth (OD<0.1) and were consequently excluded from further trials. The remaining extracts (Table 3) were selected as substrates for cultivation by utilizing nutrient broth as the basal medium to assess their effectiveness in supporting bacterial growth and pigment production.

This finding suggested that at least one AWE source had a notable impact on bacterial growth. The variation observed in biomass growth across different AWEs under similar optimized conditions could be attributed to differences in essential nutrient types and composition [44].

Pairwise comparisons in the post-hoc test revealed significant ($p < 0.05$) differences in $OD_{600\,nm}$ values for most extracts. However, no statistically significant differences ($p > 0.05$) were observed between the following pairs: tomato vs. orange, watermelon vs. cannon ball cabbage, watermelon vs. cabbage, watermelon vs. potato, and cannon ball cabbage vs. potato waste extracts, based on $OD_{600\,nm}$ measurements.

TWE demonstrated the highest $OD_{600\,nm}$ value (Table 3), making it the most effective substrate for optimizing pigment production by the potent isolate (*Exiguobacterium aurantiacum)*. Tomato peel extract is rich in carbohydrates, proteins,

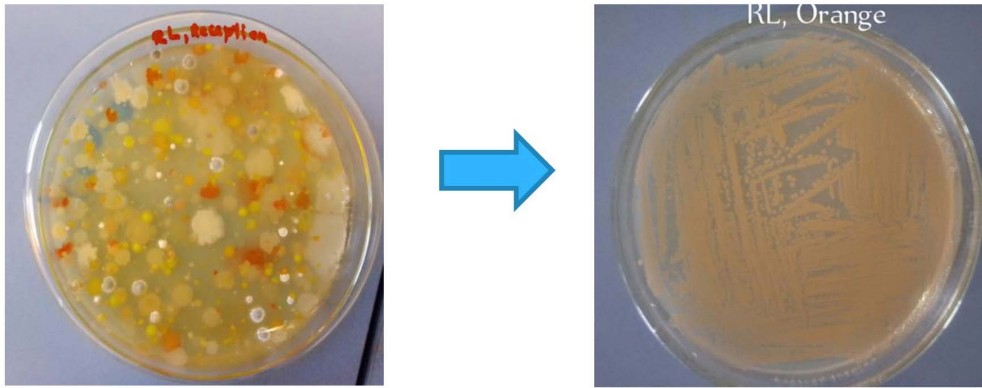

**Fig 1. Isolation of pure colony from mixture of colonies grown on nutrient agar medium using streaking method.**

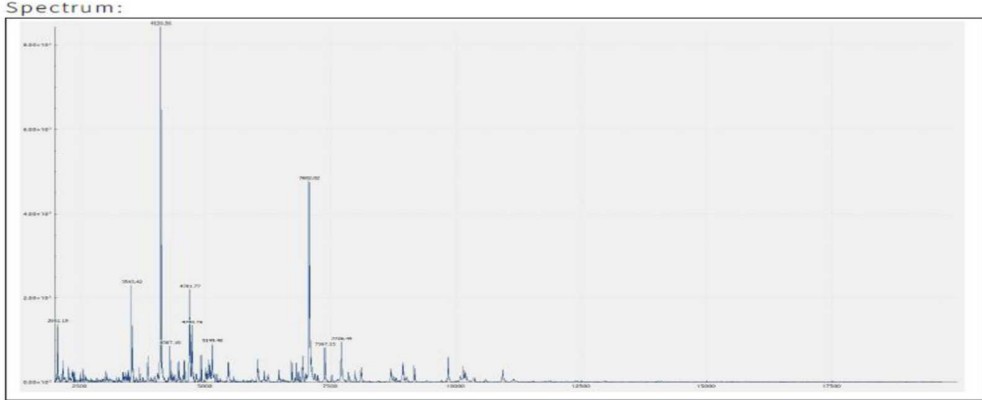

| Genus | Exiguobacterium |
|---|---|
| Species | aurantiacum |
| Score | 2.25 |
| Scoring Criteria | < 1.7 Unreliable; 1.7-2.0 Probable Genus; > 2.0 Probable Species |

Spectrum:

**Fig 2. MALDI-TOF-MS spectrum of yellowish-orange pigment producing bacteria, identified as *Exiguobacterium aurantiacum*showing distinct mass-to-charge ratio (m/z) peaks associated with bacterial proteins.**

**Table 3. One-way ANOVA of OD$_{600 nm}$ values for *Exiguobacterium aurantiacum* broth culture.**

| Group-wise OD$_{600 nm}$ measurements | | | | |
|---|---|---|---|---|
| *Substrates* | *Count* | *Sum* | *Average* | *Variance* |
| Cabbage | 3 | 1.214 | 0.405 | 3.33E-07 |
| Nutrient broth | 3 | 1.570 | 0.523 | 0.003508 |
| Onion | 3 | 0.641 | 0.214 | 2.33E-06 |
| Papaya | 3 | 0.493 | 0.164 | 1.33E-06 |
| Cannon ball cabbage | 3 | 1.304 | 0.435 | 2.33E-06 |
| Watermelon | 3 | 1.295 | 0.432 | 3.33E-05 |
| Carrot | 3 | 1.004 | 0.335 | 3.33E-07 |
| Tomato | 3 | 1.573 | 0.524 | 4.33E-06 |
| Potato | 3 | 1.385 | 0.462 | 3.33E-05 |
| Orange | 3 | 1.536 | 0.512 | 4.00E-06 |

The one-way ANOVA test (Table 4) demonstrated statistically significant differences in average OD$_{600 nm}$ values among at least two broth cultures of AWEs inoculated with *Exiguobacterium aurantiacum* ($F(9, 20) = 133.49$, $p < 0.05$).

**Table 4. ANOVA summary the experiment.**

| Source of Variation | SS | df | MS | F-value | P-value | F crit |
|---|---|---|---|---|---|---|
| Between Groups | 0.4313035 | 9 | 0.047923 | 133.49 | 7.19E-16 | 2.392814 |
| Within Groups | 0.00718 | 20 | 0.000359 | | | |
| Total | 0.4384835 | 29 | | | | |

lipids, and essential minerals including potassium, magnesium, calcium, and iron as well as vitamins C, E, and various B vitamins [45,46]. Additionally, according to [47], tomato peel contains approximately 16.9% soluble sugars, 9.21% cellulose, 10.5% hemicellulose, and 42.5% pectin, which are crucial for supporting microbial growth. These components provide a robust nutritional profile, reinforcing the extract's potential as an optimal cultivation medium.

### 3.3. Screening of process variables

Nine variables were assessed for their impact on the growth of *Exiguobacterium aurantiacum*, assessed through $OD_{600 nm}$ values (Table 5). The PBD was employed to screen and identify the most influential factors that affected bacterial culture development. This statistical approach enabled the selection of key parameters for further optimization, ensuring efficient growth and pigment production.

Analysis of the Half-Normal plot (Fig 3) and Pareto chart (Fig 4) revealed that culture agitation rate, initial culture medium pH, yeast extract, salt concentration, glucose, and inoculum size exhibited significantly ($p < 0.05$) larger effects compared to background noise. These variables were thus identified as the most influential factors affecting the growth of *Exiguobacterium aurantiacum*. Their impact highlights the necessity of precise optimization to maximize bacterial culture development and pigment production.

The half-normal plot provides a standardized visualization of process variable effects on $OD_{600 nm}$ value, where each point represents a variable, and its deviation from the straight line indicates its significance. Variables that deviate significantly from this line are identified as having substantial impacts on culture growth, highlighting critical factors for optimization.

Meanwhile, the Pareto chart ranks process variables by their influence on $OD_{600 nm}$ value in descending order, with bars representing each variable's contribution to the overall effect. This graphical representation helps prioritize key factors, enabling precise adjustments to optimize bacterial growth and pigment production. Together, these tools facilitate informed decision-making in refining cultivation conditions.

Statistical analysis using ANOVA confirmed the model's significance, with an *F*-value of 134.98 and a corresponding p-value of < 0.0001 (Table 6). The coefficient of determination ($R^2$) of 99.39% (Table 7) suggests that the selected model terms can explain nearly all variability in culture growth. Additionally, the Adjusted $R^2$ value of 98.65% accounts for the number of model terms, further reinforcing model reliability. The Predicted $R^2$ value of 96.47% indicates strong predictive

**Table 5. PBD experimental run matrix with the corresponding response, mean values of $OD_{600 nm}$.**

| Std order | Run | Assigned variables | | | | | | | | | Average $OD_{600 nm}$ |
|---|---|---|---|---|---|---|---|---|---|---|---|
| | | Incubation temperature, °C | pH | Incubation period, hr | Inoculum size, % | Inoculum Age, hr | Agitation, rpm | Yeast extract, % | Glucose, % | Salt, % | |
| 10 | 1 | 23 | 10 | 24 | 1.0 | 36 | 60 | 0.1 | 1.5 | 10 | 0.54 |
| 6 | 2 | 23 | 5 | 72 | 2.5 | 12 | 180 | 0.1 | 1.5 | 5 | 1.38 |
| 9 | 3 | 37 | 10 | 72 | 2.5 | 12 | 60 | 0.1 | 0.5 | 10 | 0.52 |
| 4 | 4 | 23 | 10 | 72 | 2.5 | 36 | 60 | 1.0 | 1.5 | 5 | 0.58 |
| 12 | 5 | 23 | 5 | 24 | 1.0 | 12 | 60 | 0.1 | 0.5 | 5 | 0.78 |
| 3 | 6 | 37 | 5 | 72 | 1.0 | 36 | 180 | 0.1 | 1.5 | 10 | 1.04 |
| 8 | 7 | 37 | 10 | 24 | 2.5 | 36 | 180 | 0.1 | 0.5 | 5 | 0.98 |
| 7 | 8 | 37 | 5 | 72 | 1.0 | 26 | 60 | 1.0 | 0.5 | 5 | 0.62 |
| 11 | 9 | 37 | 5 | 24 | 2.5 | 12 | 60 | 1.0 | 1.5 | 10 | 0.64 |
| 2 | 10 | 23 | 10 | 72 | 1.0 | 12 | 180 | 1.0 | 0.5 | 10 | 0.5 |
| 1 | 11 | 37 | 10 | 24 | 1.0 | 12 | 180 | 1.0 | 1.5 | 5 | 0.8 |
| 5 | 12 | 23 | 5 | 24 | 2.5 | 36 | 180 | 1.0 | 0.5 | 10 | 0.88 |

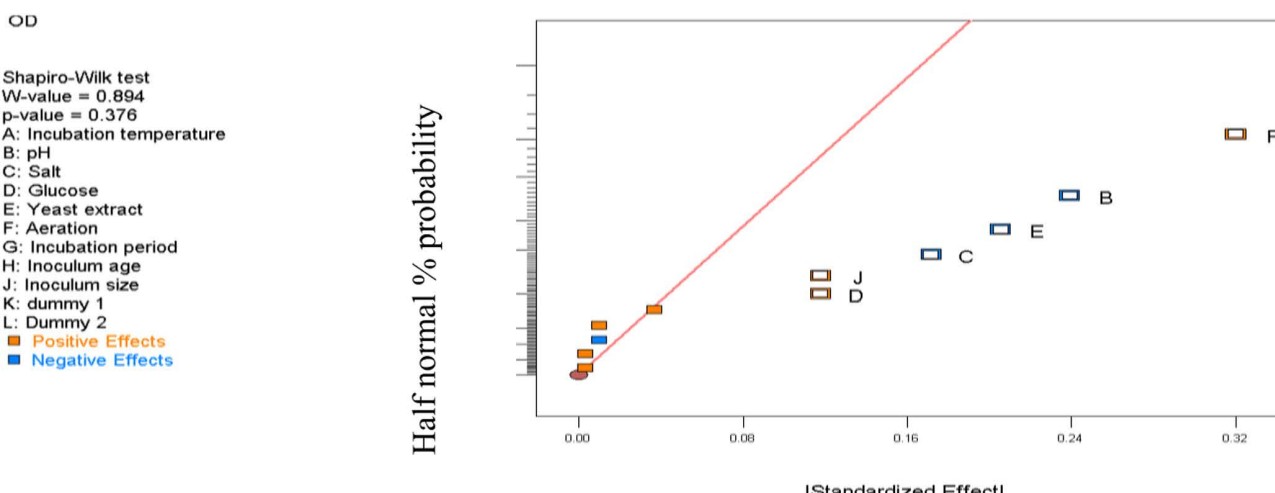

**Fig 3. Half-normal plot for the standardized effects of process variables on OD$_{600\,nm}$ value, highlighting critical factors for optimization.**

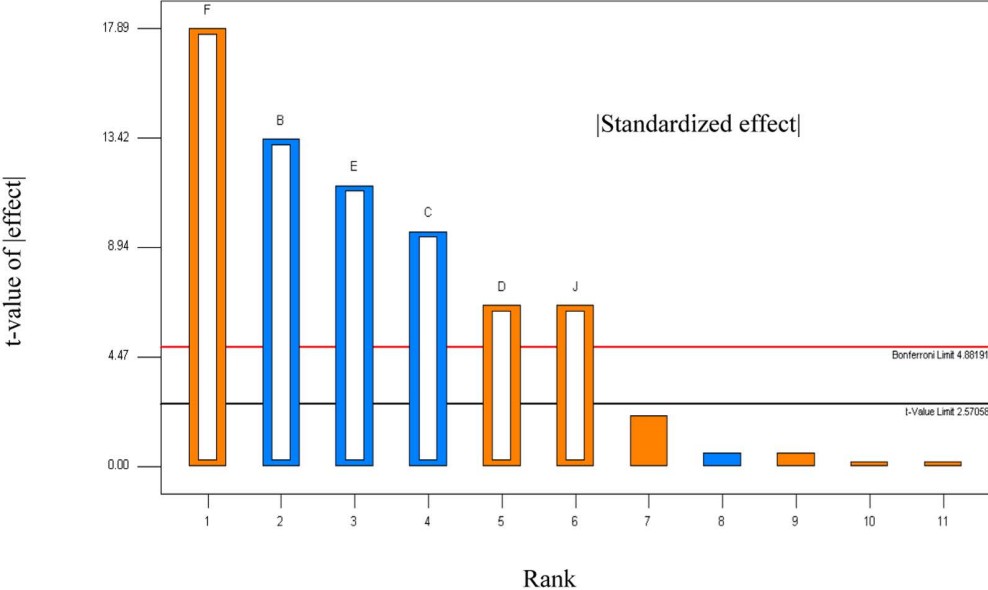

**Fig 4. Pareto chart depicting the influence of process variables on OD$_{600}$ value in descending order, bars representing each variable's contribution to the overall effect.**

accuracy for future observations under identical conditions, with good agreement between adjusted and predicted R² values, confirming a well-fitting model.

The first-order polynomial model equation was derived from ANOVA data using the coefficients presented in Table 8 as:

$$Y = 0.77 + 0.16X_1 - 0.12X_2 - 0.1X_3 \qquad (1)$$

Where Y is the OD$_{600\,nm}$ value, $X_1$ is the culture agitation rate, $X_2$ is the initial culture pH and $X_3$ is the concentration of yeast extract. This equation serves as a predictive tool for estimating culture biomass, expressed in terms of OD$_{600\,nm}$ value,

**Table 6. The ANOVA for selected factorial model.**

| Source | Sum of squares | df | Mean square | F-value | P-value Prob > F |
|---|---|---|---|---|---|
| Model | 0.76 | 6 | 0.13 | 134.98 | < 0.0001 significant |
| B-pH | 0.17 | 1 | 0.17 | 178.76 | < 0.0001 |
| C-Salt | 0.087 | 1 | 0.087 | 92.23 | 0.0002 |
| D-Glucose | 0.041 | 1 | 0.041 | 43.44 | 0.0012 |
| E-Yeast extract | 0.12 | 1 | 0.12 | 131.95 | < 0.0001 |
| F-Culture agitation | 0.30 | 1 | 0.30 | 320.04 | < 0.0001 |
| J-Inoculum size | 0.041 | 1 | 0.041 | 43.44 | 0.0012 |
| Residual | 4.700E-003 | 5 | 9.400E-004 | | |
| Cor Total | 0.77 | 11 | | | |

**Table 7. The fit statistics.**

| Std. Dev. | 0.031 | $R^2$ | 0.9939 |
|---|---|---|---|
| Mean | 0.77 | Adj $R^2$ | 0.9865 |
| C.V. % | 3.97 | Predicted $R^2$ | 0.9647 |
| PRESS | 0.027 | Adeq Precision | 36.014 |

**Table 8. The coefficients in terms of factors.**

| Factor | Coefficient Estimate | df | Standard Error | 95% CI, low | 95% CI, high | VIF |
|---|---|---|---|---|---|---|
| Intercept | 0.77 | 1 | 8.851E-003 | 0.75 | 0.79 | 1.00 |
| B-pH | − 0.12 | 1 | 8.851E-003 | − 0.14 | − 0.096 | 1.00 |
| C-Salt | − 0.085 | 1 | 8.851E-003 | − 0.11 | − 0.062 | 1.00 |
| D-Glucose | 0.058 | 1 | 8.851E-003 | 0.036 | 0.081 | 1.00 |
| E-Yeast extraction | − 0.10 | 1 | 8.851E-003 | − 0.12 | − 0.079 | 1.00 |
| F-Culture agitation | 0.16 | 1 | 8.851E-003 | 0.14 | 0.18 | 1.00 |
| J-Inoculum size | 0.058 | 1 | 8.851E-003 | 0.036 | 0.081 | 1.00 |

based on significant process variables. The generated model provides a mathematical framework for optimizing bacterial growth conditions and enhancing pigment production efficiency.

The selection of variables for further optimization was guided by their statistical significance, as determined by the *F* statistic and p-value from the ANOVA test (Table 5). Variables with the strongest impact on bacterial growth and pigment production were prioritized for refinement, ensuring that the optimization process focuses on the most influential factors.

### 3.4. Optimization of the levels of significant factors

The face-centered central composite design (CCD) experiment investigated how agitation speed (rpm), pH, and yeast extract concentration influenced bacterial growth and pigment production. The response variables, $OD_{600\,nm}$ value ($Y_1$), culture biomass ($Y_2$), and pigment yield ($Y_3$) were measured at the stationary growth phase.

Each experimental run in Table 9 represents the mean response values under different conditions, allowing the interaction effects of significant factors to be assessed, influencing the $OD_{600\,nm}$ value as a measure of bacterial growth. $OD_{600\,nm}$ values ($Y_1$) were used to measure bacterial growth, reflecting cell density under different experimental conditions. Biomass and pigment yields were collected at the stationary phase where maximum cell density was reached and quantified using the gravimetric method.

**Table 9. Face-centered CCD experiment design matrices and response values.**

| Std | Run | Factors | | | Responses | | |
|---|---|---|---|---|---|---|---|
| | | Agitation (rpm) | pH | Yeast extract (%) | Mean $OD_{600\,nm}$ value ($Y_1$) | Mean culture biomass (g/L) ($Y_2$) | Mean pigment yield (g/L) ($Y_3$) |
| 11 | 1 | 60 | 7.5 | 0.55 | 0.65 | 3.6 | 0.82 |
| 13 | 2 | 120 | 7.5 | 0.1 | 0.8 | 4.7 | 0.84 |
| 2 | 3 | 60 | 10 | 0.1 | 0.71 | 4.1 | 0.85 |
| 9 | 4 | 120 | 5 | 0.55 | 0.78 | 4.3 | 0.8 |
| 12 | 5 | 180 | 7.5 | 0.55 | 0.6 | 3.4 | 0.76 |
| 7 | 6 | 180 | 5 | 1 | 0.58 | 3.68 | 0.74 |
| 16 | 7 | 120 | 7.5 | 0.55 | 0.74 | 4.12 | 0.8 |
| 5 | 8 | 60 | 5 | 1 | 0.77 | 4.12 | 0.74 |
| 8 | 9 | 180 | 10 | 1 | 0.89 | 4.2 | 0.88 |
| 1 | 10 | 60 | 5 | 0.1 | 0.88 | 4.62 | 1.06 |
| 19 | 11 | 120 | 7.5 | 0.55 | 0.74 | 4.12 | 0.8 |
| 4 | 12 | 180 | 10 | 0.1 | 0.83 | 4.0 | 0.73 |
| 18 | 13 | 120 | 7.5 | 0.55 | 0.74 | 4.2 | 0.8 |
| 14 | 14 | 120 | 7.5 | 1 | 0.78 | 4.5 | 0.76 |
| 3 | 15 | 180 | 5 | 0.1 | 0.71 | 3.82 | 0.72 |
| 17 | 16 | 120 | 7.5 | 0.55 | 0.74 | 4.2 | 0.82 |
| 20 | 17 | 120 | 7.5 | 0.55 | 0.72 | 4.1 | 0.78 |
| 6 | 18 | 60 | 10 | 1 | 0.78 | 3.92 | 0.65 |
| 10 | 19 | 120 | 10 | 0.55 | 0.84 | 4.2 | 0.78 |
| 15 | 20 | 120 | 7.5 | 0.55 | 0.72 | 4.12 | 0.8 |

The results indicate that moderate agitation (120 rpm), a balanced pH (around 7.5), and optimized yeast extract concentration contribute to improved bacterial density and pigment synthesis. The highest pigment yield was observed at 60 rpm, pH 5, and 0.1% yeast extract, suggesting nutrient availability and environmental conditions strongly influence microbial performance.

Table 10 summarizes the ANOVA results for $OD_{600\,nm}$ value ($Y_1$), culture biomass ($Y_2$), and pigment yield ($Y_3$), emphasizing the impact of key independent factors-pH (A), agitation speed (B), and yeast extract concentration (C). These variables were found to significantly ($p < 0.05$) influenced bacterial growth and pigment production, guiding optimization strategies to enhance overall efficiency.

The polynomial equation coefficients derived from the ANOVA (Table 10) were used to predict response variables, including $OD_{600\,nm}$ value ($Y_1$), culture biomass ($Y_2$), and pigment yield ($Y_3$). The regression equations below represent the mathematical relationships between the independent factors and the predicted outcomes, allowing for precise modeling of bacterial growth and pigment production under optimized conditions.

$$Y_1 = 0.7318 + 0.033X_1 - 0.018X_2 - 0.013X_3 + 0.0738X_1X_2 + 0.0463X_1X_3 - 0.0037X_2X_3 + 0.0805X_1^2 - 0.1045X_2^2 + 0.0605X_3^2 \quad (2)$$

$$Y_2 = 4.14 - 0.012X_1 - 0.126X_2 - 0.082X_3 + 0.1775X_1X_2 + 0.0825X_1X_3 + 0.0925X_2X_3 + 0.1045X_1^2 - 0.6455X_2^2 + 0.4545X_3^2 \quad (3)$$

$$Y_3 = 0.7965 - 0.017X_1 - 0.029X_2 - 0.043X_3 + 0.0562X_1X_2 + 0.0312X_1X_3 \quad (4)$$

Where $X_1$ is culture agitation rate, $X_2$ is initial culture pH, and $X_3$ is concentration of yeast extract.

The Intercept Values for $OD_{600\,nm}$, culture biomass, and pigment yield are 0.7318, 4.14, and 0.7965, respectively; representing the baseline response values when all factors are at their central levels. Analysis of main effects showed pH (A) has a positive effect on $OD_{600\,nm}$ but a slight negative effect on biomass and pigment yield. Whereas, agitation (B) and yeast extract (C) showed negative impact across all responses, suggesting higher agitation may reduce biomass and pigment production and excessive yeast extract concentration may limit growth and pigment yield.

Interaction effects revealed that AB (pH × agitation), AC (pH × yeast extract), and BC (agitation × yeast extract) exhibited positive interaction effects, suggesting that these factor combinations contribute to improved bacterial growth and pigment synthesis. On the other hand, quadratic effects, pH (A²) and yeast extract (C²) positively influenced responses, suggesting an optimal level improved biomass and pigment yield. On the contrary, agitation (B²) showed a strong negative effect, particularly on biomass (−0.6455), indicating excessive agitation negatively impacts bacterial density.

Model significance analysis indicated strong statistical significance, showing that the model reliably predicts responses. P-values (< 0.0001) confirmed the high significance of the model, and R² values (~0.99) indicated that the model explains nearly all the variabilities in the data and Adjusted R² and predicted R² values confirmed a strong correlation between experimental data and model predictions. The lack of fit p-values were also not significant (p > 0.05) relative to the pure error, suggesting that the models were statistically accurate.

The results highlighted the influence of pH, agitation speed, and yeast extract concentration on bacterial growth and pigment production. Moderate agitation, balanced pH, and optimized yeast extract levels improved microbial performance. Additionally, interaction effects between factors played crucial role in maximizing yield, offering valuable insights for industrial applications.

### 3.5. Analysis of the effects of the selected culture conditions on response variables

The influence of various significant factors on culture growth and pigment production was examined across different ranges, with outcomes expressed in $OD_{600\,nm}$ value, culture biomass (g/L), and pigment yield (g/L). The ANOVA test results for these response variables, summarized in Table 10, provide valuable insights into the statistical significance of each factor.

**Table 10. The ANOVA results for responses.**

| Factor | Mean $OD_{600\,nm}$ value ($Y_1$) | Mean culture biomass (g/L) ($Y_2$) | Mean pigment yield (g/L) ($Y_3$) |
|---|---|---|---|
| Intercept | 0.7318 | 4.14 | 0.7965 |
| A-pH | 0.0330 | −0.0120 | −0.0170 |
| B-Agitation | −0.0180 | −0.1260 | −0.0290 |
| C-Yeast extract | −0.0130 | −0.0820 | −0.0430 |
| AB | 0.0738 | 0.1775 | 0.0562 |
| AC | 0.0463 | 0.0825 | 0.0312 |
| BC | −0.0037 | 0.0925 | 0.0862 |
| A² | 0.0805 | 0.1045 | |
| B² | −0.1045 | −0.6455 | |
| C² | 0.0605 | 0.4545 | |
| Model F-value | 179.14 | 130.58 | 216.10 |
| Model P-value | < 0.0001 | < 0.0001 | < 0.0001 |
| Lack of Fit P-value | 0.8145 | 0.6927 | 0.9185 |
| R² | 0.9938 | 0.9916 | 0.9901 |
| Adjusted R² | 0.9883 | 0.9840 | 0.9855 |
| Predicted R² | 0.9799 | 0.9678 | 0.9841 |

**3.5.1. Effects on OD value.** The $OD_{600\,nm}$ value measurements revealed that bacterial cell concentration in the nutritious medium (TWE) was significantly affected by culture agitation rate, displaying linear, quadratic, and interaction effects with initial medium pH ($p < 0.0001$) (Table 11). However, yeast extract concentration did not show a statistically significant effect ($p > 0.05$). These findings underscore the importance of optimizing agitation rate and medium pH to enhance bacterial growth while highlighting that yeast extract may not be a critical determinant in this process.

Culture agitation plays a vital role in enhancing oxygen transfer, ensuring uniform nutrient distribution, and maintaining consistent temperature throughout the growth medium. A study demonstrated that increasing mixing speed significantly improves bacterial growth rates [48].

Another critical factor influencing culture growth was pH, which exhibited significant linear, quadratic, and interaction effects with yeast extract concentration ($p < 0.0001$). Maintaining optimal pH levels is essential for promoting enzymatic activity, maximizing bacterial growth, and enhancing fermentation efficiency [49]. These findings highlight the importance of carefully adjusting agitation speed and pH conditions to achieve optimal bacterial development and pigment production.

The response surface graph (Fig 5) visually illustrates the interaction effects among culture agitation, yeast extract concentration, and initial culture pH on bacterial growth. This graphical representation helps identify optimal conditions by highlighting how variations in these factors impact biomass development. The response surfaces provide insights into the best combinations to maximize bacterial growth and pigment production.

These graphs illustrate how culture agitation rate, yeast extract concentration, and pH interact to influence various output variables. By adjusting two independent factors within experimental ranges while holding the third constant at the center point, the response patterns became clearer.

Fig 5, panel (A) highlights the relationship between agitation speed (rpm) and pH on $OD_{600\,nm}$ value, revealing an optimal $OD_{600\,nm}$ value of 0.85 at an agitation rate of around 140 rpm and a pH of 10. This aligns with previous findings that *Exiguobacterium aurantiacum* thrives within a pH range of 5–11 [50]. Furthermore, the optimal agitation rate identified falls within the recommended 130–150 rpm range for bacterial cultures [51], reinforcing the reliability of these growth conditions.

The response surface graph in Fig 5, panel B, illustrates the linear, quadratic, and interaction effects of yeast extract concentration and initial medium pH on $OD_{600\,nm}$ value. The convex curvature suggests a minimum point where their combination results in the lowest $OD_{600\,nm}$ value.

**Table 11. The ANOVA for Quadratic model, Response: $OD_{600}$ nm.**

| Source | Sum of Squares | df | Mean Square | *F*-value | p-value | |
|---|---|---|---|---|---|---|
| Model | 0.1226 | 9 | 0.0136 | 179.14 | < 0.0001 | significant |
| A-pH | 0.0109 | 1 | 0.0109 | 143.16 | < 0.0001 | |
| B-Agitation | 0.0032 | 1 | 0.0032 | 42.59 | < 0.0001 | |
| C-Yeast extract | 0.0017 | 1 | 0.0017 | 22.22 | 0.0008 | |
| AB | 0.0435 | 1 | 0.0435 | 572.02 | < 0.0001 | |
| AC | 0.0171 | 1 | 0.0171 | 224.96 | < 0.0001 | |
| BC | 0.0001 | 1 | 0.0001 | 1.48 | 0.2519 | |
| A² | 0.0178 | 1 | 0.0178 | 234.01 | < 0.0001 | |
| B² | 0.0301 | 1 | 0.0301 | 395.13 | < 0.0001 | |
| C² | 0.0101 | 1 | 0.0101 | 132.13 | < 0.0001 | |
| Residual | 0.0008 | 10 | 0.0001 | | | |
| Lack of Fit | 0.0002 | 5 | 0.0000 | 0.4263 | 0.8145 | not significant |
| Pure Error | 0.0005 | 5 | 0.0001 | | | |
| Corrected Total | 0.1234 | 19 | | | | |

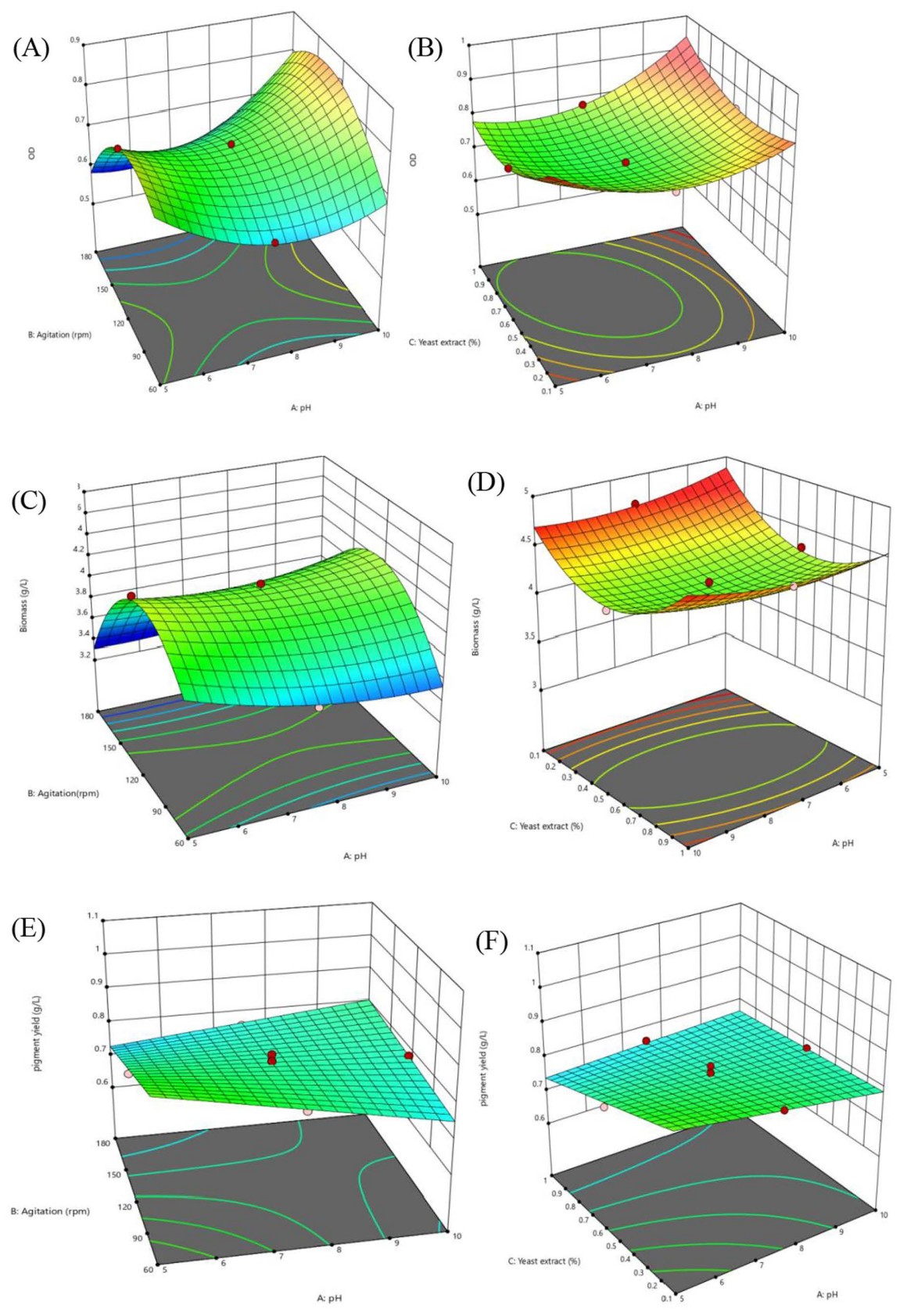

**Fig 5. Response surface optimization graphs, panels (A)-(F).** Panel (A) shows the relationship between agitation speed (rpm) and pH on OD value. Panel B illustrates effects of yeast extract concentration and initial medium pH on OD value. Panels C and D elucidate the combined effects of agitation versus pH and yeast extract versus pH on culture biomass yield, and panels E and F demonstrates how pigment yield is influenced by agitation rate, pH, and yeast extract concentration.

The highest $OD_{600\ nm}$ values were recorded at both the highest yeast extract and pH levels, as well as at the lowest yeast extract concentration paired with extreme pH values, demonstrating the bacterial strain's pH tolerance. This aligns with findings from [52], which indicate that while optimal yeast extract concentration can vary based on experimental conditions, a concentration of around 0.7% has been found effective for microbial growth.

**3.5.2. Effects on culture biomass.** Culture agitation demonstrated a significant impact on bacterial growth at linear, quadratic, and interaction levels, particularly with pH and yeast extract concentration (Table 12). Other key contributors to culture biomass included the linear and quadratic terms of yeast extract ($p < 0.0001$), the quadratic term of medium pH ($p < 0.0015$), and the interaction between yeast extract and pH ($p < 0.002$). However, the linear effect of initial medium pH was not statistically significant ($p > 0.05$).

Bacterial growth rate and final biomass influence pigment production. Hence, optimizing these factors can enhance overall yield. Studies suggest that fine-tuning agitation, pH, and nutrient concentrations can substantially improve biomass accumulation, ultimately leading to efficient pigment production [53,54].

Ensuring an optimal balance of nutrient availability, pH stability, and proper agitation rates is fundamental for maximizing bacterial biomass production [55]. Since microbial growth is highly sensitive to nutrient composition and environmental conditions, fine-tuning these parameters can significantly enhance biomass yield and pigment production efficiency [49].

Optimizing bacterial cultivation involves tailoring these factors to the species specific requirements, creating an environment that fosters rapid growth while supporting metabolic activity crucial for pigment biosynthesis.

The response surface optimization graph (Fig 5, panels C and D) illustrates the combined effects of agitation versus pH and yeast extract versus pH on culture biomass yield. The data show that biomass production increases with rising pH levels, while both excessively high and low agitation rates result in reduced biomass yield (Fig 5, panel C).

Excessive agitation can induce shear stress, potentially damaging cells and lowering biomass output [55]. Conversely, insufficient agitation reduces oxygen transfer rates, creates nutrient gradients, and promotes cellular clumping, slowing

**Table 12. The ANOVA for Quadratic model, Response: Biomass.**

| Source | Sum of Squares | df | Mean Square | *F*-value | p-value | Remark |
|---|---|---|---|---|---|---|
| Model | 1.89 | 9 | 0.2103 | 130.58 | < 0.0001 | significant |
| A-pH | 0.0014 | 1 | 0.0014 | 0.8940 | 0.3667 | |
| B-Agitation | 0.1588 | 1 | 0.1588 | 98.56 | < 0.0001 | |
| C-Yeast extract | 0.0672 | 1 | 0.0672 | 41.74 | < 0.0001 | |
| AB | 0.2520 | 1 | 0.2520 | 156.47 | < 0.0001 | |
| AC | 0.0545 | 1 | 0.0545 | 33.80 | 0.0002 | |
| BC | 0.0684 | 1 | 0.0684 | 42.49 | < 0.0001 | |
| A² | 0.0301 | 1 | 0.0301 | 18.66 | 0.0015 | |
| B² | 1.15 | 1 | 1.15 | 711.24 | < 0.0001 | |
| C² | 0.5682 | 1 | 0.5682 | 352.73 | < 0.0001 | |
| Residual | 0.0161 | 10 | 0.0016 | | | |
| Lack of Fit | 0.0062 | 5 | 0.0012 | 0.6216 | 0.6927 | not significant |
| Pure Error | 0.0099 | 5 | 0.0020 | | | |
| Corrected Total | 1.91 | 19 | | | | |

metabolic activity and ultimately decreasing biomass production [55–57]. These insights emphasize the need for precise regulation of agitation speed and pH to optimize bacterial growth and pigment synthesis efficiency.

The response surface optimization graph (Fig 5, panel D) illustrates the interactive effects of yeast extract concentration and pH on culture biomass yield. A consistent upward trend was observed across all the pH ranges, with peak biomass production occurring at lower yeast extract concentrations, followed by a slight increase at higher levels.

Lower nitrogen source concentrations help prevent nitrogen toxicity, which can hinder microbial growth, thereby fostering a balanced environment that enables efficient nutrient utilization for biomass production [52]. The adaptability of *Exiguobacterium aurantiacum* to a broad pH spectrum (5–11) allows the bacterium to sustain metabolic activity and growth under varying conditions, enhancing its biomass yield [50,58]. These findings reinforce the importance of optimizing nutrient composition and pH conditions to maximize bacterial productivity.

**3.5.3. Effects on pigment yield.** Table 13 reveals that pigment yield was significantly affected by initial culture pH, agitation rate, and yeast extract concentration at both linear and two-factor interaction (2FI) levels (p < 0.0001). These results underscore the importance of fine-tuning these variables to maximize pigment production efficiency.

The response surface optimization graph (Fig 5, panels E and F) demonstrates how pigment yield is influenced by agitation rate, pH, and yeast extract concentration. At lower agitation and yeast extract levels, pigment production significantly (p < 0.05) increased as pH decreased. This trend may be linked to a stress-induced metabolic response, where reduced pH and nutrient availability trigger enhanced secondary metabolite synthesis that boosted pigment yield. Additionally, lower agitation minimizes shear stress on bacterial cells, creating a more stable environment conducive to pigment biosynthesis [36,59].

## 3.6. Verification tests using TWE as cultivation medium and pigment extraction

The pre-optimization cultivation experiments in 150 mL TWE yielded 0.65 g/L of crude pigment from a culture biomass of 4.20 g/L, with an $OD_{600\,nm}$ value of 0.60 at the stationary growth phase, based on averaged measurements. For validation, forty-six different solutions were generated by the software, ranked according to their desirability values, which quantify how well each solution meets optimization criteria.

The highest-ranked solution, with a desirability of 98.8%, was chosen to test the predictive model (Table 14). The selected optimal conditions were a pH of 5, an agitation rate of 65 rpm, and a yeast extract concentration of 0.1%. These parameters offer a refined approach for maximizing pigment yield and bacterial growth.

To ensure the reliability and reproducibility of the predictive model, culture cultivation was conducted in triplicate under optimized conditions in 150 mL of TWE. During the experiment, bacterial growth dynamics was closely monitored by

**Table 13. The ANOVA for 2FI model, Response: pigment yield.**

| Source | Sum of Squares | df | Mean Square | *F*-value | p-value | Remark |
|---|---|---|---|---|---|---|
| Model | 0.1224 | 6 | 0.0204 | 216.10 | < 0.0001 | significant |
| A-pH | 0.0029 | 1 | 0.0029 | 30.61 | < 0.0001 | |
| B-Agitation | 0.0084 | 1 | 0.0084 | 89.07 | < 0.0001 | |
| C-Yeast extract | 0.0185 | 1 | 0.0185 | 195.82 | < 0.0001 | |
| AB | 0.0253 | 1 | 0.0253 | 268.08 | < 0.0001 | |
| AC | 0.0078 | 1 | 0.0078 | 82.74 | < 0.0001 | |
| BC | 0.0595 | 1 | 0.0595 | 630.27 | < 0.0001 | |
| Residual | 0.0012 | 13 | 0.0001 | | | |
| Lack of Fit | 0.0004 | 8 | 0.0001 | 0.3340 | 0.9185 | not significant |
| Pure Error | 0.0008 | 5 | 0.0002 | | | |
| Corrected Total | 0.1237 | 19 | | | | |

**Table 14. The solutions generated by the software for verification experiment.**

| Number | pH | Aeration | Yeast extract | OD$_{600\,nm}$ | Biomass | pigment yield | Desirability | |
|--------|------|----------|---------------|----------------|---------|---------------|--------------|----------|
| 1 | 5.000 | 64.969 | 0.100 | 0.892 | 4.700 | 1.045 | 0.988 | Selected |
| 2 | 5.000 | 65.835 | 0.100 | 0.893 | 4.711 | 1.043 | 0.986 | |
| 3 | 5.027 | 65.474 | 0.100 | 0.890 | 4.702 | 1.043 | 0.986 | |
| 4 | 5.000 | 65.609 | 0.103 | 0.892 | 4.700 | 1.042 | 0.985 | |
| 5 | 5.000 | 62.375 | 0.100 | 0.887 | 4.665 | 1.052 | 0.982 | |

measuring OD$_{600\,nm}$ (Fig 6). These measurements provided valuable insights into growth trends, confirming whether the selected parameters effectively enhanced bacterial biomass and pigment production.

The growth kinetics of *Exiguobacterium aurantiacum* under optimized conditions, using TWE as the cultivation substrate demonstrated its effectiveness in promoting bacterial growth. The growth curve (OD$_{600\,nm}$ measurements over time) captured distinct bacterial phases, confirming TWE as a sustainable medium.

Optimal growth was observed on the 7$^{th}$ day, when the culture reached the stationary phase, possibly due to environmental stress according to [60]. Biomass was harvested via centrifugation on the 8$^{th}$ day to recover the pigment, yielding 4.73 g/L of biomass and 0.96 g/L of crude pigment (Table 15). The experimental verification showed a minor 0.63% error in biomass concentration prediction, likely due to slight model noise and challenges in maintaining precise conditions. Overall, the validation confirmed strong alignment between the predicted and actual results, reinforcing the model's reliability.

Comparison of pigment yield highlights how substrate composition and culture conditions impact bacterial pigment production. The yield of 0.96 g/L exceeds the 180 µg/g biomass reported for *Chryseobacterium* sp. using feather meal [61]. However, the yield falls short of previous results obtained from *Micrococcus luteus* cultivated in orange waste extract as a substrate [62], likely due to differences in nutrient composition and optimization conditions [63].

These variations underscore the importance of tailoring cultivation strategies based on substrate properties, bacterial strain requirements, and environmental factors to maximize pigment synthesis.

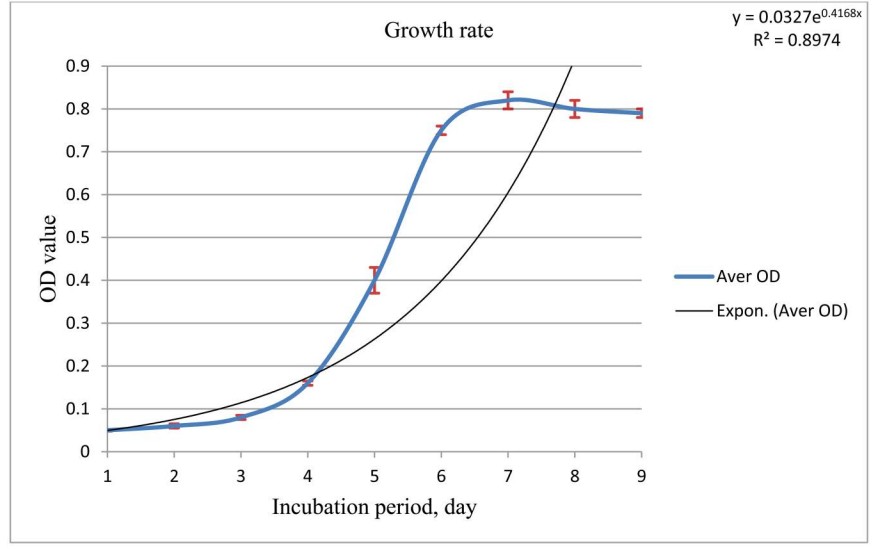

**Fig 6. Smooth line graph showing OD$_{600}$ variation over time of *Exiguobacterium aurantiacum* at optimized process conditions using TWE as cultivation substrate with standard deviation represented as error bars around the line.**

**Table 15. The predicted and experimental response values using TWE as substrate at optimized conditions.**

| Optimum condition | Coded levels | | Actual levels | |
|---|---|---|---|---|
| Culture agitation (rpm) | − 0.92 | | 65 | |
| Initial culture pH | −1.00 | | 5.0 | |
| Yeast extract (%) | −1.00 | | 0.1 | |
| Response | Un-optimized value | Predicted value | Optimized value | Optimization efficiency (%) |
| $OD_{600\ nm}$ value | 0.60 | 0.892 | 0.877 ± 0.025 | 31.6 |
| Cell biomass (g/L) | 4.20 | 4.700 | 4.733 ± 0.076 | 11.3 |
| Pigment yield (g/L) | 0.65 | 1.045 | 0.957 ± 0.060 | 32.1 |

### 3.7. Pigment characterization

**3.7.1. FTIR spectrum analysis.** The spectroscopy analysis (Fig 7) presents the IR spectrum of the pigmented compound extracted from *Exiguobacterium aurantiacum*, providing insights into its functional groups and molecular structure. Infrared (IR) spectroscopy detects specific bond vibrations, allowing the identification of characteristic absorption peaks associated with various functional groups within the pigment molecule.

The spectrum displays characteristic absorption peaks corresponding to various functional groups present in the compound, which provide insights into the molecular structure and chemical composition of the pigment, aiding in its identification and analysis.

Th IR spectrum showed bands of functional groups with peaks at 3270.86, 2921.81, 2852.96, 1738.65, and 1629.69 cm$^{-1}$. The fingerprint regions with different absorption bands provide information about the chemical composition of the pigment extracted from the isolate, *Exiguobacterium aurantiacum*.

The weak and broad absorption peak at 3270.87.76 cm$^{-1}$ is associated with O-H stretching vibrations, characteristic of hydroxyl groups. The second and third short and narrow peaks at 2921.81 and 2852.96 cm$^{-1}$ indicate C-H stretching

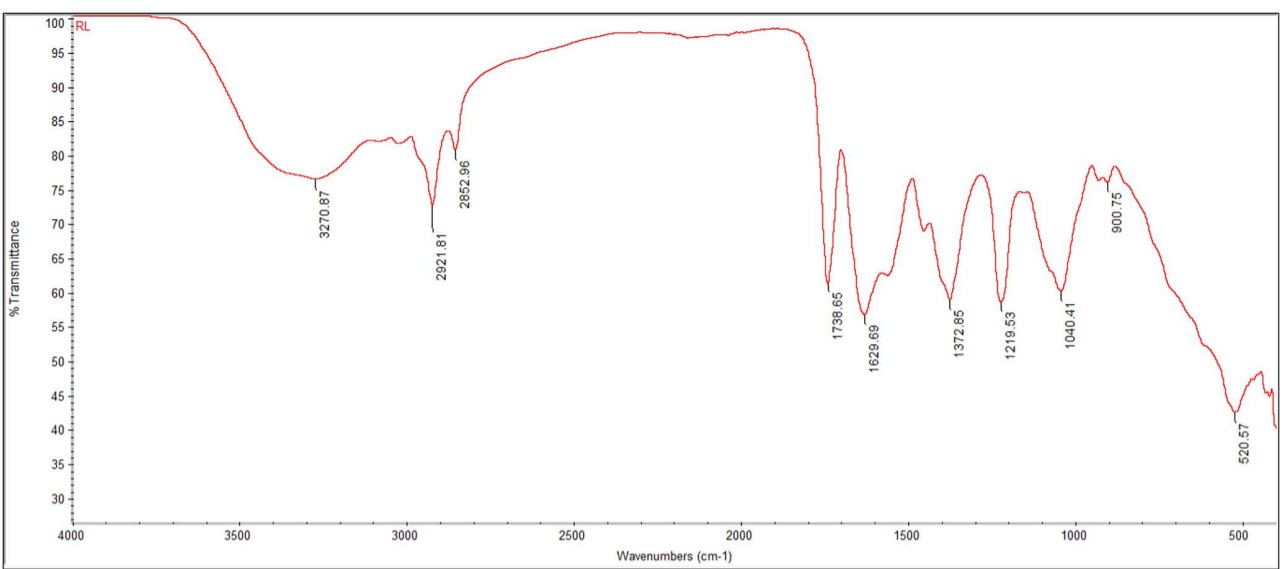

**Fig 7. IR spectrum of pigmented compound extracted from *Exiguobacterium aurantiacum*, depicting characteristic absorption peaks corresponding to various functional groups present in the compound.**

vibrations commonly found in alkanes. The narrow peak at 1738.65 cm$^{-1}$ is characteristic of C=O stretching vibrations, typically found in carbonyl groups, and the peak at 1629.69 cm$^{-1}$ is associated with C=C stretching vibrations found in alkenes.

These Vibrations help identify different functional groups in molecules by their characteristic absorption bands. The functional group frequency analysis revealed that the IR spectrum of the yellowish-orange pigment extracted from *Exiguobacterium aurantiacum* indicates the presence of hydroxyl, hydrocarbon, olefinic, and carbonyl groups, which are among the functional groups present in carotenoid compounds [64].

**3.7.2. UV-visible spectroscopy analysis.** The spectral scanning of the methanolic extract of the yellowish-orange pigment, conducted over wavelengths ranging from 350–750 nm (Fig 8), provides crucial insights into its optical properties. By analyzing absorption peaks within this range, the study aims to determine the characteristic absorbance profile of the pigment, aiding in its identification and potential application.

The UV-visible spectrophotometric analysis of the crude pigment extracted from *Exiguobacterium aurantiacum* revealed distinct absorption peaks, confirming the presence of chromophores within the pigment. These peaks provide valuable insights into the molecular structure and composition of the extracted compound [54,65].

The characteristic absorbance peak at 467 nm falls within the 400–550 nm range typical of carotenoid compounds, a crucial pigment group responsible for yellow to orange-red coloration [66,67]. Carotenoids exhibit strong absorption in this range due to π-electron delocalization within their conjugated system, allowing efficient light absorption in the UV-visible spectrum [68].

**3.7.3. LC-MS analysis.** The chromatogram (Fig 9) offers a detailed visual representation of the chromatographic separation and mass spectrometric detection of the extracted pigment, showcasing distinct peaks corresponding to various carotenoid molecules. These peaks provide key insights into the composition, purity, and molecular identity of the pigment, helping to confirm its classification.

The LC-MS analysis provided a detailed characterization of carotenoid compounds extracted from *Exiguobacterium aurantiacum*. The Total Ion Current (TIC) plot (Fig 9, panel A) illustrates the ion intensity detected over time, highlighting distinct peaks corresponding to compounds eluting from the chromatography column at different intervals. These peaks represent the molecular components analyzed by the mass spectrometer.

The mass spectra (Fig 9, panels B and C) reveal peaks corresponding to the mass-to-charge ratios (m/z) of the detected carotenoid molecules, offering insights into their molecular weights and fragmentation patterns. The LC-MS investigation confirmed the presence of carotenoid compounds, specifically 1,4-Naphthalenedione,

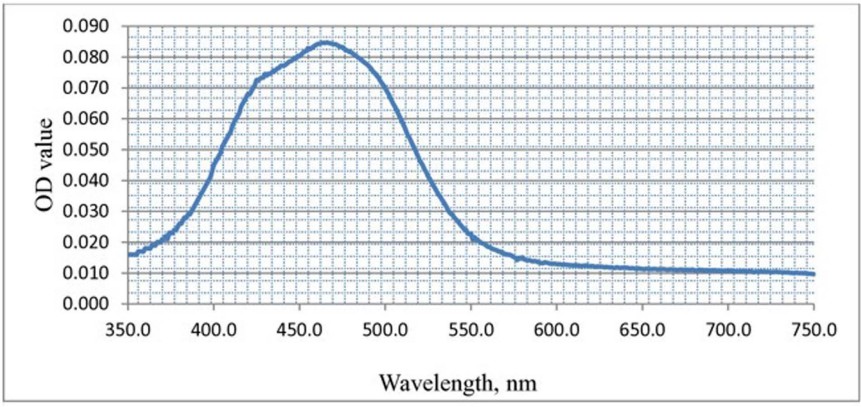

**Fig 8. Absorption spectrum of pigment extracted from *Exiguobacterium aurantiacum* scanned from 350-750 nm using UV-visible spectrophotometer.**

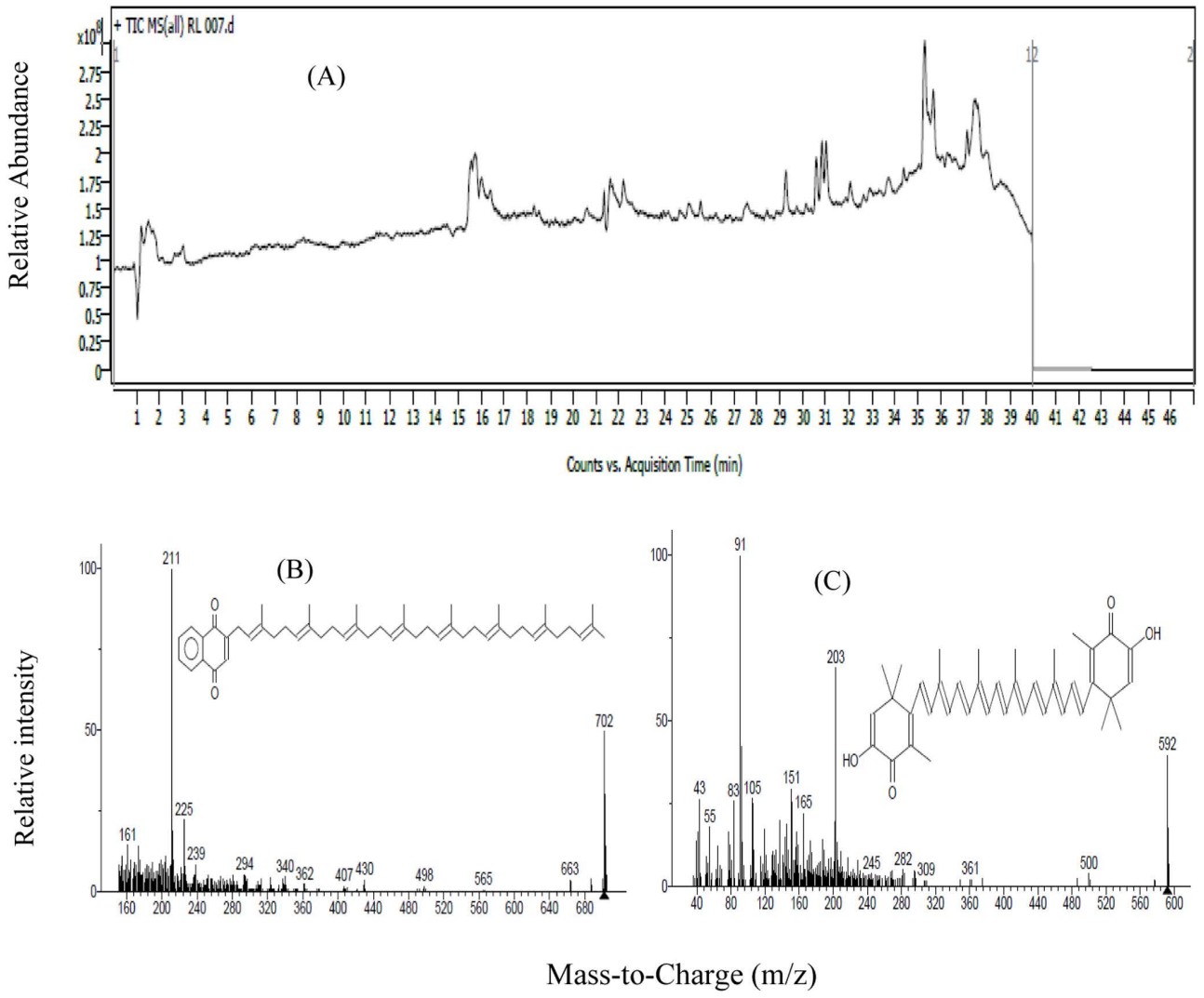

**Fig 9. LC-MS analysis of Carotenoid compounds extracted from *Exiguobacterium aurantiacum*.** Panel A illustrates the ion intensity detected over time, panels B and C reveal peaks corresponding to the mass-to-charge ratios (m/z) of the detected carotenoid molecules.

2-(3,7,11,15,19,23,27,31-octamethyl-2,6,10,14,18,22,26,30-dotriacontaoctaenyl)-(all-E) and 2,5-Cyclohexadien-1-one,3,3'-(3,7,11,15-tetramethyl-1,3,5,7,9,11,13,15,17-octadecanonaene-1,18-diyl)-bis[6-hydroxy-2,4,4-trimethyl].

These compounds belong to the naphthoquinone and polyene classes, known for their intense coloration due to conjugated double bonds that enable absorption of visible light. Their structural properties contribute to the yellowish-orange hue of the pigment extracted from *E. aurantiacum*, reinforcing their classification within the carotenoid group.

## 4. Conclusion

Eco-friendly natural pigment production and extraction was successfully carried. The study effectively demonstrated the potential of utilizing agro-waste extracts, especially tomato waste as a cost-effective and nutritious substrate for the cultivation of pigment-producing bacterial isolate, *Exiguobacterium aurantiacum*. The successful optimization using statistical methods yielded 0.96 g/L of pigment from 4.3 g of bacterial biomass, reinforcing the efficiency of this approach. By

transforming agro-waste into valuable products, this research not only enhances microbial pigment production but also promotes waste valorization, reducing environmental impact and supporting sustainability efforts. The findings provide a convincing case for further exploration of agro-waste applications in biotechnological industries.

## Acknowledgments

The authors are thankful to Adama Science and Technology University, Adama Public Health Research and Referral Laboratory Center, Adama, and Food and Drug Authority and Wudasie Diagnostic Center, Addis Ababa, Ethiopia, for allowing to use their laboratory facilities.

List of the individual authors and their affiliations:

1. Birhanu Zeleke, Department of Applied Biology, School of Applied Natural Science, Adama Science and Technology University, Adama, Oromia, Ethiopia.
2. Diriba Muleta, Environmental Biotechnology Unit, Institute of Biotechnology, Addis Ababa University, Addis Ababa, Ethiopia.
3. Hunduma Dinka, Department of Applied Biology, School of Applied Natural Science, Adama Science and Technology University, Adama, Oromia, Ethiopia.
4. Dereje Tsegaye, Department of Applied Chemistry, School of Applied Natural Science, Adama Science and Technology University, Adama, Oromia, Ethiopia.
5. Jemal Hassen, Adama Public Health Research and Referral Laboratory Center, Adama, Oromia, Ethiopia.

## Author contributions

**Conceptualization:** Birhanu Zeleke, Dr Diriba Muleta, Hunduma Dinka, Dereje Tsegaye.

**Data curation:** Birhanu Zeleke, Dr Diriba Muleta, Hunduma Dinka, Dereje Tsegaye.

**Formal analysis:** Birhanu Zeleke, Jemal Hassen.

**Funding acquisition:** Hunduma Dinka, Dereje Tsegaye.

**Investigation:** Birhanu Zeleke.

**Methodology:** Birhanu Zeleke, Jemal Hassen.

**Project administration:** Dr Diriba Muleta, Hunduma Dinka, Dereje Tsegaye.

**Resources:** Dr Diriba Muleta, Hunduma Dinka, Dereje Tsegaye, Jemal Hassen.

**Software:** Birhanu Zeleke.

**Supervision:** Dr Diriba Muleta, Hunduma Dinka, Dereje Tsegaye.

**Validation:** Birhanu Zeleke, Dr Diriba Muleta, Hunduma Dinka, Dereje Tsegaye, Jemal Hassen.

**Visualization:** Birhanu Zeleke, Dr Diriba Muleta, Hunduma Dinka, Dereje Tsegaye, Jemal Hassen.

**Writing – original draft:** Birhanu Zeleke.

**Writing – review & editing:** Birhanu Zeleke, Dr Diriba Muleta, Hunduma Dinka, Dereje Tsegaye, Jemal Hassen.

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
