## [Editor Report · Decision Letter 0]

Dear Dr. Muleta,

Thank you for submitting your manuscript to PLOS ONE. After careful consideration, we feel that it has merit but does not fully meet PLOS ONE’s publication criteria as it currently stands. Therefore, we invite you to submit a revised version of the manuscript that addresses the points raised during the review process.

We look forward to receiving your revised manuscript.

Kind regards,

Guru Prasad Srinivasan

Academic Editor

PLOS ONE

Journal Requirements:

2. group authorship (wait for confirmation)

One of the noted authors is a group or consortium Birhanu Zeleke, Hunduma Dink, Dereje Tsegaye, Jemal Hassen. In addition to naming the author group, please list the individual authors and affiliations within this group in the acknowledgments section of your manuscript. Please also indicate clearly a lead author for this group along with a contact email address.
---

## [Author Response · Author response to Decision Letter 1]

18 Nov 2024

We really appreciate the given comments that critical fine-tune the Ms. All the suggested comments are incorporated.

---

## [Editor Report · Decision Letter 1]

Dear Dr. Muleta,

Thank you for submitting your manuscript to PLOS ONE. After careful consideration, we feel that it has merit but does not fully meet PLOS ONE’s publication criteria as it currently stands. Therefore, we invite you to submit a revised version of the manuscript that addresses the points raised during the review process.

**ACADEMIC EDITOR:**

The manuscript describes about the use of natural pigment using TWE but the conclusion needs to be comprehensivePlease justify how the Use of two statistical methods will help when this study is taken for Scale up levelsIn the abstract it is finally given that this study was done using AWE but in contrast the conclusion part says TWE is better - Please justifyThe manuscript should be checked for language corrections 

Please submit your revised manuscript by Feb 17 2025 11:59PM. If you will need more time than this to complete your revisions, please reply to this message or contact the journal office at plosone@plos.org . A rebuttal letter that responds to each point raised by the academic editor and reviewer(s). You should upload this letter as a separate file labeled 'Response to Reviewers'.A marked-up copy of your manuscript that highlights changes made to the original version. You should upload this as a separate file labeled 'Revised Manuscript with Track Changes'.An unmarked version of your revised paper without tracked changes. You should upload this as a separate file labeled 'Manuscript'.

We look forward to receiving your revised manuscript.

Kind regards,

Guru Prasad Srinivasan

Academic Editor

PLOS ONE
---

## [Author Response · Author response to Decision Letter 2]

3 Feb 2025

We are highly grateful to Editor and the anonymous reviewers for critical comments that greatly fine-tune our Ms. We have tried our level best to incorporate all the suggested changes as indicated point-by-point in the file named 'Response to Reviewers.'

---

## [Decision Letter · Decision Letter 2]

Dear Dr. Muleta,

Thank you for submitting your manuscript to PLOS ONE. After careful consideration, we feel that it has merit but does not fully meet PLOS ONE’s publication criteria as it currently stands. Therefore, we invite you to submit a revised version of the manuscript that addresses the points raised during the review process.

We look forward to receiving your revised manuscript.

Kind regards,

Guadalupe Virginia Nevárez-Moorillón, Ph.D.

Academic Editor

PLOS ONE

Reviewers' comments:

Reviewer's Responses to Questions

**Comments to the Author**

Reviewer #1: (No Response)

Reviewer #2: All comments have been addressed

2. Is the manuscript technically sound, and do the data support the conclusions?

Reviewer #1: Partly

Reviewer #2: Yes

3. Has the statistical analysis been performed appropriately and rigorously?

Reviewer #1: Yes

Reviewer #2: Yes

4. Have the authors made all data underlying the findings in their manuscript fully available?

Reviewer #1: Yes

Reviewer #2: Yes

5. Is the manuscript presented in an intelligible fashion and written in standard English?

Reviewer #1: No

Reviewer #2: Yes

Reviewer #1: The manuscript describes optimization of pigment production by a bacterial strain. Authors adopted statistical tools to improve the pigment production. Apparently it is a revised version but authors still need to correct various aspects.

1. Language of the article needs improvement. There are several misleading statements such as in the line 70.

2. Abstract mainly describes methods. It should be improved to state key findings, as well.

3. In text citation should be corrected

4. Whenever write OD, mention wavelength in subscript else it will remain unclear to the readers

5. L152-154: Write about inoculum preparation and its standardization

6. Section 2.6: There isn’t any characterization given here. It is merely identification and elucidation of its structure

7. 3.1: Describe why this isolate was selected

8. Improve presentation of the table 3

9. Table 8 and 9: Why OD values are given here when it is considered as a raw data

10. Table 13: Why run 5 was not selected when it gives better results

11. Fig. 6: Standard deviation is missing

12. L521: State which figure has been referred here

13. Table 14: What is optimization efficiency?

Reviewer #2: The manuscript presents an interesting approach to optimizing pigment production from a bacterial source using agro-waste, addressing a relevant need for sustainable pigment alternatives. The use of statistical methods to optimize culture conditions is generally well-executed, but some aspects require clarification and improvement.

Specific Comments:

Statistical Design:

The combination of Plackett-Burman Design (PBD) for screening and Response Surface Methodology (RSM) for optimization is a sound strategy. PBD is appropriate for initially identifying significant variables from a larger pool, and RSM (specifically the CCD used here) is well-suited for modeling and optimizing the selected factors.

However, the justification for selecting the specific variables for the PBD could be strengthened. Why these nine variables initially? Are they based on prior knowledge or literature?

The levels chosen for the independent variables in both the PBD and RSM designs appear reasonable, but it would be beneficial to briefly explain the rationale behind selecting those specific ranges.

**Do you want your identity to be public for this peer review?** For information about this choice, including consent withdrawal, please see our Privacy Policy

Reviewer #1: No

Reviewer #2: No

---

## [Author Response · Author response to Decision Letter 3]

26 May 2025

All the suggested concerns were comprehensively addressed (cf the uploaded document as response to Reviewers).

---

## [Editor Report · Decision Letter 3]

Enhancing pigment production by a chromogenic bacterium (Exiguobacterium aurantiacum) using tomato waste extract: A statistical approach

PONE-D-24-46181R3

Dear Dr. Muleta,

We’re pleased to inform you that your manuscript has been judged scientifically suitable for publication and will be formally accepted for publication once it meets all outstanding technical requirements.

Kind regards,

Guadalupe Virginia Nevárez-Moorillón, Ph.D.

Academic Editor

PLOS ONE
---

## [Editor Report · Acceptance letter]

PONE-D-24-46181R3

PLOS ONE

Dear Dr. Muleta,

I'm pleased to inform you that your manuscript has been deemed suitable for publication in PLOS ONE. Congratulations! Your manuscript is now being handed over to our production team.

Kind regards,

on behalf of

Dr. Guadalupe Virginia Nevárez-Moorillón

Academic Editor

PLOS ONE